# Saliva as a Biological Fluid in SARS-CoV-2 Detection

**DOI:** 10.3390/diagnostics14090922

**Published:** 2024-04-29

**Authors:** Emily Thalia Teixeira Silva, Fabiana Barcelos Furtado, Rosana Antunes da Silveira, Karen Ingrid Tasca, Cristiane Nonato Silva, Amanda Thais Godoy, Leonardo Nazario de Moraes, Michelle Venancio Hong, Camila Gonçalves Alves, Rafael Plana Simões, Agatha Mayume Silva Kubo, Carlos Magno Castelo Branco Fortaleza, Maria Cristina Pereira-Lima, Guilherme Targino Valente, Rejane Maria Tommasini Grotto

**Affiliations:** 1Laboratory of Applied Biotechnology, Medical School, São Paulo State University (Unesp), Botucatu 18618-689, Brazil; emily.tt.silva@unesp.br (E.T.T.S.); fabiana.furtado@unesp.br (F.B.F.); rosana.silveira@unesp.br (R.A.d.S.); cristiane.nnt@gmail.com (C.N.S.); amanda.godoy@unesp.br (A.T.G.); leonardo.nazario@unesp.br (L.N.d.M.); agatha.kubo@unesp.br (A.M.S.K.); 2Department of Infectious Diseases, Dermatology, Imaging Diagnosis, and Radiotherapy, Medical School, São Paulo State University (Unesp), Botucatu 18618-689, Brazil; karen.i.tascaa@unesp.br (K.I.T.); michelle.hong@unesp.br (M.V.H.); goncalves.alves@unesp.br (C.G.A.); carlos.fortaleza@unesp.br (C.M.C.B.F.); 3Department of Bioprocess and Biotechnology, School of Agriculture, São Paulo State University (Unesp), Botucatu 18618-689, Brazil; rafael.simoes@unesp.br; 4Department of Neurology, Psychology and Psychiatry, Medical School, São Paulo State University (Unesp), Botucatu 18618-689, Brazil; maria.cristina@unesp.br

**Keywords:** COVID-19, diagnosis, RT-qPCR, cycle threshold

## Abstract

Background: The polymerase chain reaction of upper respiratory tract swab samples was established as the gold standard procedure for diagnosing SARS-CoV-2 during the COVID pandemic. However, saliva collection has attracted attention as an alternative diagnostic collection method. The goal of this study was to compare the use of saliva and nasopharyngeal swab (NPS) samples for the detection of SARS-CoV-2. Methods: Ninety-nine paired samples were evaluated for the detection of SARS-CoV-2 by saliva and swab for a qualitative diagnosis and quantitative comparison of viral particles. Furthermore, the detection limits for each sample collection technique were determined. The cycle threshold (C_T_) values of the saliva samples, the vaccination status, and the financial costs associated with each collection technique were compared. Results: The results showed qualitative equivalence in diagnosis (96.96%) comparing saliva and swab collection, although there was low quantitative agreement. Furthermore, the detection limit test demonstrated equivalence for both collection methods. We did not observe a statistically significant association between C_T_ values and vaccination status, indicating that the vaccine had no influence on viral load at diagnosis. Finally, we observed that the use of saliva incurs lower financial costs and requires less use of plastic materials, making it more sustainable. Conclusions: These findings support the adoption of saliva collection as a feasible and sustainable alternative to the diagnosis of COVID-19.

## 1. Introduction

The COVID-19 pandemic prompted researchers to explore alternative diagnostic methods to detect SARS-CoV-2, leading to the investigation of the feasibility of using saliva samples for the detection of its viral RNA. Saliva collection has several advantages over nasopharyngeal swab (NPS), including non-invasiveness, self-collection, and reduced risk of contamination of healthcare professionals [1]. Despite these advantages, the NPS method is still considered the gold standard method for the diagnosis of COVID-19 [2].

Tests using saliva and nasopharyngeal swab (NPS) collection presented equivalent results for the detection of SARS-CoV-2 [3,4], although further investigation is needed to confirm this hypothesis. The use of saliva exhibits higher sensitivity in symptomatic patients; in these circumstances, saliva can be considered diagnostically equivalent to the nasopharyngeal swab [5]. The use of saliva can outperform swab collection methods when comparing cycle threshold (C_T_) values [6,7]. Furthermore, saliva use is suitable for monitoring asymptomatic individuals, while the swab method shows better diagnostic performance [8,9]. The meta-analysis evaluated the impact of the sampling site on the accuracy of RT-PCR diagnostic tests in the diagnosis of SARS-CoV-2 infection after the emergence of the omicron variant. Finally, the tests using saliva samples did not achieve results comparable to the nasopharyngeal swab in terms of specificity and sensitivity [10].

Given the conflicting data presented here, additional studies are needed to validate an accurate diagnosis of COVID-19 through saliva samples. Here, we conducted a paired qualitative assessment of the diagnosis and a quantitative assessment of viral particles in NPS and saliva samples using quantitative reverse transcription polymerase chain reaction (RT-qPCR). Furthermore, the determination of the detection limits was carried out for each collection technique. We also compared C_T_ values and vaccination status and the financial costs related to each technique. Our study provides important information on the use of saliva as a collection method for the diagnosis of SARS-CoV-2 and its advantages over NPS.

## 2. Materials and Methods

### 2.1. Samples Description

An initial cohort of 99 positive saliva samples was obtained in a COVID-19 monitoring programme that attended patients and staff of a hospital; 69 individuals out of 99 cases were symptomatic (~70%) and 30 individuals asymptomatic (~30%). A retrospective analysis from 4 January 2021 to 31 January 2022 was performed using data from the Applied Biotechnology Laboratory at the Hospital das Clínicas da Faculdade de Medicina de Botucatu (HCFMB). All positive individuals identified by a saliva sample were convocated for collection of a nasopharyngeal swab (NPS) sample to confirm the initial detection. The NPS samples included in this study were collected up to 24 h after saliva sampling.

Both types of samples (saliva and NPS) were collected in the hospital setting. The collection using swabs was carried out by specialized professionals from HCFMB. The samples were collected with a rayon swab from both the nostrils and the oropharynx, which were placed into a conical tube containing 3 mL of 0.9% saline solution. Saliva samples were collected in sterile microtubes by patients, who fasted for two hours before collection. Then, approximately 1 mL of saliva was transferred to a microtube with 500 µL of 0.9% saline. Samples were stored at 4 °C until use.

### 2.2. Ethical Approval

This study was approved by the institution’s Research Ethics Committee (Faculdade de Medicina de Botucatu—FMB; CAAE: 49984321.9.0000.5411).

### 2.3. Viral RNA Extraction

For RNA extraction, 200 µL of the sample was previously lysed in 200 µL of in-house lysis solution (5 M guanidine thiocyanate, 100 mM Tris HCl, 7 mM EDTA, and 20% Triton X-100). Then, the RNA was isolated/purified using an in-house protocol based on the Solid Phase Reversible Immobilization (SPRI) technique, which consisted of homogenizing the sample previously lysed in SPRI buffer (20% polyethylene glycol 8000 (PEG-800), 3.65 M NaCl, 0.05% Tween 20, 1 mM sodium citrate and 0.1% GE Sera-Mag Magnetic SpeedBeads Carboxylated, cat. GE65152105050250). Magnetic beads were recovered using the Extracta 96 automated extractor (Loccus, Cotia, São Paulo, Brasil) and washed twice in 80% ethanol. Finally, the RNA was solubilized in 60 µL of ultrapure water and the beads were discarded.

### 2.4. SARS-CoV-2 RT-qPCR Assay

The real-time reverse transcription polymerase chain reaction (RT-qPCR) was performed using the Allplex™ SARS-CoV-2 Assay (Seegene, Seoul, Republic of Korea), with modifications. The RT-qPCR mixture contained 1 X real time one-step buffer, 5 µL de 2019-nCoV MOM, 2 µL real-time one-step enzyme, and 5 µL of viral RNA extracted from saliva in a final volume of 25 µL. The cycling conditions were 20 min at 50 °C, 15 min at 95 °C, followed by 45 cycles of 15 s at 94 °C and 1 min at 58 °C. Amplification was performed using the Applied Biosystems™ 7500 Real-Time PCR System (Thermo Fisher Scientific, Waltham, MA, USA) or the CFX96 Touch Real-Time PCR Detection System (BioRad, Hercules, CA, USA). We used the baseline and threshold parameters provided by the manufacturer for each detection system. Our analysis criteria considered a positive sample based on a C_T_ (cycle threshold) value <40 detected in at least 2 of the 3 viral genes analyzed (envelope gene (E gene), nucleocapsid protein gene (N gene) and RNA-dependent RNA polymerase (RdRP gene)). The C_T_ value of the E gene was used for the comparative analysis between saliva and NPS samples.

### 2.5. Limit of Detection (LoD) Determination

The limit of detection (LoD) test determined the lowest detectable concentration (copies of RNA/mL) in which ≥95% of the replicates were positive (true positives). For this purpose, we selected retrospective saliva and NPS samples positive for SARS-CoV-2 with C_T_ value < 22. A pool of 30 positive saliva samples, and a pool of 30 positive NPS samples from 50 µL of each sample were formed. Another RNA extraction was performed in triplicate followed by RT-qPCR to verify the C_T_ value after thawing.

An initial range-finding study was performed using a 10-fold dilution series of pooled SARS-CoV-2 positive samples. After identifying the point of loss in sensibility, 2-fold serial dilutions were used to confirm the lowest concentration, at which 95% (19/20) of the replicates were positive.

To classify a replicate as positive, we followed our diagnostic criteria previously mentioned, requiring the detection of at least two viral genes with a C_T_ value < 40.

The estimation of the SARS-CoV-2 RNA copy number of the E gene was based on the use of a serial diluted RNA previously quantified in our laboratory by droplet digital PCR (ddPCR). RNA extraction was performed from a clinical specimen expanded in cell culture.

### 2.6. Comparison of C_T_ Values and Vaccination Status

C_T_ values and vaccination status were compared in an independent set of exclusive saliva samples (159 cases). Vaccination status was obtained from the municipality’s epidemiological surveillance systems (E-sus, SIVEP-Gripe and Vacivida). We considered individuals with up to 6 days of symptoms. The number of days was defined by independently assessing the mean C_T_ of each day and comparing the mean values among groups. Statistical differences were not found until the sixth day. C_T_s were classified into 3 groups: values < 20, from 20 to 30, and >30. Regarding the status of vaccination, the individuals were classified as unvaccinated or vaccinated with at least 3 doses: they were separated into groups that received the same type of vaccine technology (viral vector vaccine, chemically inactivated virus, RNA vaccine) or with at least two different types of technology.

### 2.7. Cost Comparison

The costs of molecular diagnosis of SARS-CoV-2 from saliva and NPS were compared considering personal protective equipment (PPE) and materials used for collection, PPE for the receiving and processing teams, reagents, consumables, and labor for carrying out all stages.

### 2.8. Statistical Analysis

The Wilcoxon Signed Rank test was applied to compare the median C_T_ values between NPS and saliva using the R software v 4.3.0 with the RcmdrMisc package v 2.8-0 [11]. Bland–Altman analysis was performed using STATA software v 14.0 (StataCorp LLC, College Station, TX, USA). The chi-square test was performed using Sigma Plot 11.0 software. The significance level for the tests was established at α = 0.05.

## 3. Results

The overview of our main findings is presented in Figure 1.

### 3.1. Detection of SARS-CoV-2 in Saliva and NPS Samples

Saliva and NPS collection were performed for ninety-nine individuals. SARS-CoV-2 was detected in 96.96% (96/99) of NPS samples collected up to 24 h after saliva collection. A total of three samples had viral detection only in saliva; two out of three had C_T_ > 35.

The differences in the median C_T_ values for the E gene were statistically significant (*p* < 0.001): 19.19 [(IQR) 16.71–22.58] and 26.29 [(IQR) 24.01–29.91], respectively, for NPS and saliva (Figure 2).

### 3.2. Bland–Altman Agreement Analysis

Bland–Altman analysis was performed to assess agreement between C_T_ values of saliva and NPS samples. A plot was constructed for a total of 96 paired samples. We observed a bias of 6.69 (SD = 5.78, limits of agreement = −4.64 to 18.03) for saliva C_T_ values compared to NPS C_T_ values (Figure 3), showing weak agreement between the two measurements.

### 3.3. Limit of Detection (LoD)

LoD was determined to verify the lowest concentration that produced at least 95% of positive replicates. Following the criteria in our routine laboratory tests, a replicate that presented amplification of minimal number of two viral genes was considered a positive sample.

Saliva and NPS showed similar LoD for the E gene with 15.48 and 14.37 copies/µL, respectively (Table 1).

### 3.4. Cycle Threshold versus Status Vaccinal of 159 Saliva Samples

There was no statistically significant association between the three ranges of C_T_ values assessed and vaccination status (*p* = 0.289) (Table 2). Vaccination status was categorized considering manufacturing technology, since different vaccines were available.

Based on the findings, the administration of the vaccine does not appear to impact the viral load of individuals who are later infected by SARS-CoV-2. Furthermore, taking vaccines based on the same manufacturing technology, or different vaccines, did not affect the viral load.

### 3.5. Cost Comparison Analysis: Swab vs. Saliva

This study compared the costs of performing COVID-19 diagnostic tests using real-time PCR for swab and saliva collections (Table 3). We evaluated the costs of PPE for the collection team, materials for collection, PPE for the receiving and processing teams, reagents, consumables, and human resources for all stages.

## 4. Discussion

In terms of qualitative diagnosis, the detection of SARS-CoV-2 was successfully confirmed in 96 out of 99 patients using saliva or swab collection up to a 24 h interval between collections. For the three samples in which the swab did not show a positive result, two were within the detection limits (C_T_ = 35). Therefore, the percentage of confirmation of the diagnosis by saliva or swab collection was 96.96%, with only one inconsistent result. This observation underscores the understanding that within the limits of detection, which was observed for the other two samples, the results can sometimes be positive or negative. Such variability can be attributed to the low viral load present in the sample and equipment limitations associated with detection limits. Another aspect to consider is the interval between collections, as the viral load tends to decrease over time; however, this decrease depends on the adaptive immunity of each individual [12].

Considering the quantitative viral load results, the Bland–Altman and Wilcoxon tests showed weak agreement between the collection methods and a significant difference between the C_T_ medians, respectively. The diagnosis of COVID-19 does not depend on a precise quantification of viral load but rather on a qualitative assessment of the presence or absence of SARS-CoV-2 genetic material. Therefore, saliva collection is possible for diagnostic purposes, as our results also show a high qualitative correspondence (96.96%).

Our data corroborate previous findings regarding a similar qualitative agreement between the two types of collection (98%). However, unlike our results, quantitative C_T_s data showed a strong and significant correlation [13]: this same range of agreement was also previously observed [14,15]. Other studies also showed degrees of equivalence, but with different results [16,17,18,19]. Our findings are consistent with previous research indicating a comparable qualitative agreement between saliva and swab collections, although there are variations in the quantitative data between the studies.

SARS-CoV-2 enters the body primarily through the respiratory tract, specifically through the mucous membranes. The virus gains entry by binding to the angiotensin-converting enzyme 2 receptor on host cells [20]. The virus was detected in the salivary glands of COVID-19 patients, indicating that these glands can serve as a reservoir for the virus [21]. Postmortem biopsies conducted in fatal cases of COVID-19 revealed the presence of SARS-CoV-2 in salivary gland samples, with positive RT-qPCR results and ultrastructural evidence of viral particles within salivary gland cells. This finding underscores the potential for SARS-CoV-2 to infect and replicate in salivary gland cells, contributing to the presence of the virus in saliva and highlighting the role of salivary glands in viral dissemination and contamination [21]. Altogether, the literature and the results found here reaffirm saliva as a suitable biological fluid for the detection of SARS-CoV-2.

The limit of detection analyzed here reported that 95% of LoD for the Allplex™ SARS-CoV-2 Assay in our cohort was similar for saliva and NPS samples. A similar LoD value with the AllPlex kit was previously reported [22]: data were calculated from the detection of in vitro transcribed RNA from the SARS-CoV-2 E gene. Thus, our study showed that saliva use did not make any difference in terms of the minimum amount of SARS-CoV-2 RNA required for detection and could be used as a replacement for NPS.

Analysis of the cycle threshold versus vaccination status showed that the ability to detect SARS-CoV-2 through saliva appears to be unaffected by the vaccination process. In fact, vaccination does not appear to affect the potential infectivity of an individual when infected with the Delta variant [23]. Furthermore, people can be infected with SARS-CoV-2 and have a high viral load regardless of vaccination [24]. Taking into account the literature and the results observed here, it seems that vaccination status does not result in an increase in false negatives in diagnosing SARS-CoV-2.

Based on the cost analysis, we observed that the swab collection costs are approximately 75.15% higher than the saliva collection. The highest impact on these values is the costs of the professional who performs the collection and nonbiodegradable inputs, such as the swab and plastic conical tubes, which also involves concerns about the waste generated. Therefore, the cost comparison between saliva and swab collection for the diagnosis of SARS-CoV-2 indicates that saliva collection may offer potential cost savings. However, it is important to consider that the specific cost comparison between the two collection methods may vary depending on the testing facility, location, and other factors. Furthermore, the results of this study support the idea that saliva collection requires less plastic materials, making it more sustainable compared to the swab collection method. Saliva collection is more comfortable for patients and a safer option for healthcare professionals.

## 5. Conclusions

This study demonstrated qualitative agreement between saliva and swab collections for the detection of SARS-CoV-2, with a confirmation rate of 96.96%. Although there were discrepancies in quantitative viral load measurements between the two methods, the main focus of COVID-19 diagnosis is qualitative assessment rather than precise quantification. Saliva collection showed a detection limit comparable to that of nasopharyngeal swabs. Furthermore, vaccination status did not significantly affect the ability to detect the virus via saliva, reinforcing its reliability in diagnostic protocols. These findings support the adoption of saliva collection as a feasible and sustainable alternative to the diagnosis of COVID-19.

## Figures and Tables

**Figure 1 diagnostics-14-00922-f001:**
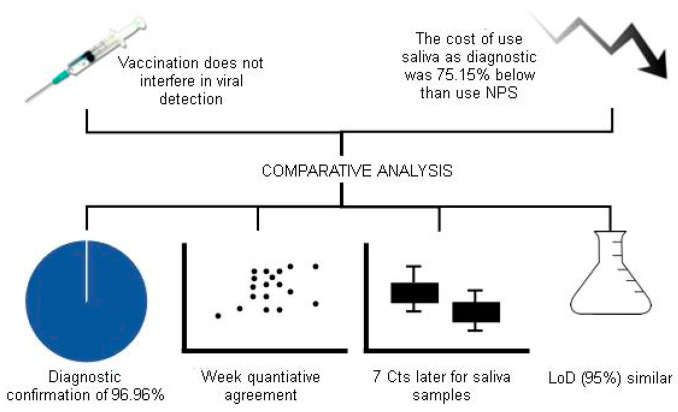
Graphical abstract summarizing our findings.

**Figure 2 diagnostics-14-00922-f002:**
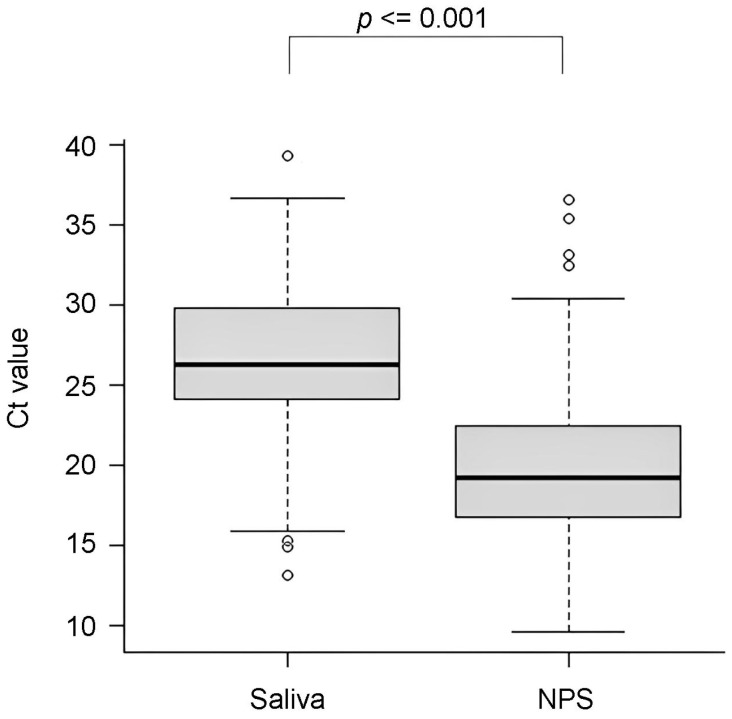
Median cycle threshold (C_T_) values detected for saliva and NPS samples (*p*-value from the Wilcoxon Signed Rank test; the dark lines inside the boxes are the median, and the lower and upper edges represent the first and third quartiles, respectively).

**Figure 3 diagnostics-14-00922-f003:**
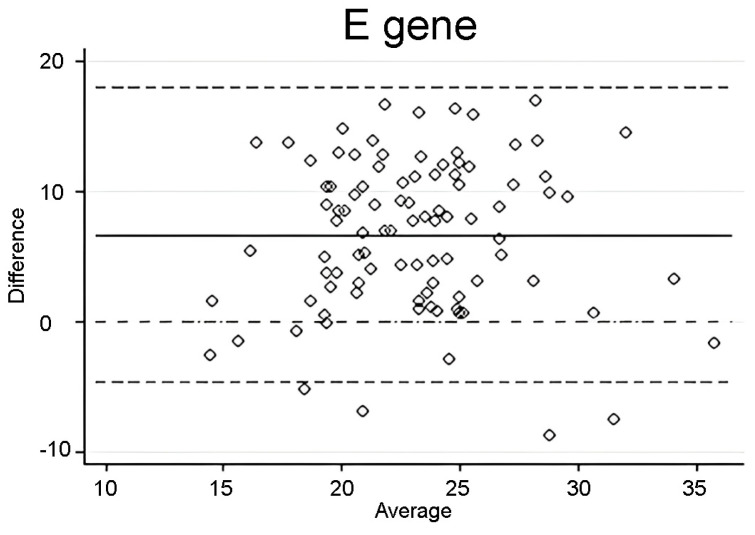
Bland-Altman assessment was conducted on C_T_ values obtained for the SARS-CoV-2 E gene from saliva and nasopharyngeal swab samples collected within a 24 h timeframe. The dashed lines indicate the limits of agreement, while the solid line represents the mean difference between the measurements.

**Table 1 diagnostics-14-00922-t001:** Limit of detection (LoD) for saliva and NPS samples.

	Saliva	NPS ^1^
Positives/Total	19/20	20/20
RNA concentration ^2^	15.48	14.37
Average of C_T_ ^3^	36.3	36.6
Standard deviation of C_T_	0.85	1.35

^1^ Nasopharyngeal swab; ^2^ concentration in copies of RNA/µL; ^3^ C_T_ value of E gene.

**Table 2 diagnostics-14-00922-t002:** Relationship between vaccination status and ranges of cycle threshold (C_T_) values of saliva samples.

Vaccinal Status/C_T_	<20	Between 20 and 30	>30
No vaccination	3	7	1
Three doses with the same vaccine type	1	4	3
At least two different vaccine types	13	89	38

The characteristics that define the contingency table are not significantly related (*p* = 0.289).

**Table 3 diagnostics-14-00922-t003:** Comparison of approximate costs between SARS-CoV-2 detection tests for saliva and swab collections.

Collection Stage (598 Samples/Month) ^&^		
Material/Human resources	Saliva	Swab
Conical Tubes (unit)	-	USD 93.23
Swab Rayon (unit)	-	USD 205.70
Microtubes (unit)	USD 44.73	-
Saline	USD 0.81	USD 4.88
Nursing Technician Professional	-	USD 625.00 × 4 ^£^ = 2500
Total	USD 45.48	USD 2803.88
Total/Sample	USD 0.075	USD 4.68
Separation and processing stage (30 samples/day) *		
Material	Saliva	Swab
Tips 200 µL	USD 3.21	-
Pasteur Pipette 3 mL	-	USD 0.78
Cylindrical Tube 4 mL	-	USD 0.33
Full Sleeve Disposable Apron (unit)	USD 0.65	USD 0.65
Disposable Nitrile Gloves (pair)	USD 0.045	USD 0.045
N95 Disposable Mask (unit)	USD 0.18	USD 0.18
Disposable Caps (unit)	USD 0.016	USD 0.016
Total	USD 4.11	USD 2.02
Total/Sample	USD 0.13	USD 0.06
Viral RNA extraction step (same method for saliva e swab) 96 samples		
Material		Cost
Tire Encapsulation Magnetics		USD 9.39
Plate Deep Well 2.2 mL viral RNA extraction Loccus Extracta 96		USD 5.46
Tips 1000 µL		USD 12.21
Plate Sealing Film (Non-Optical)		USD 1.46
Magnetics Beads		USD 10.10
Sodium Chloride		USD 0.18
Triton x-100		USD 0.36
Guanidine Thiocyanate		USD 3.96
Polyethyleneglycol		USD 1.80
Total		USD 44.96
Total/Sample		USD 0.46
RT-qPCR (same method for saliva e swab) 96 samples		
Material		Cost
Kit Allplex™ SARS-CoV-2 Assay		USD 499.19 ^#^
Tips 0.5–10 µL		USD 10.33
96-well plates for real time PCR-100 µL wells		USD 3.74
PCR Plate Sealing Film		USD 1.87
Full Sleeve Disposable Apron (unit)		USD 0.65
Disposable Nitrile Gloves (pair)		USD 0.24
N95 Disposable Mask (unit)		USD 0.045
Disposable Caps (unit)		USD 0.016
Total		USD 516.05
Total/Sample		USD 5.37
Totals Costs		
Stage	Saliva	Swab
Collect	USD 0.075	USD 4.68
Separation and processing	USD 0.13	USD 0.06
Viral RNA extraction	USD 0.46	USD 0.46
RT-qPCR	USD 5.37	USD 5.37
Total/Sample	USD 6.05	USD 10.59

^&^ Monthly average of swab samples collected by HC-FMB in 2021; ^£^ in 2021, two nursing technicians were hired to meet HC-FMB’s swab collection demands; * average number of swab samples received per day by LBA—Molecular Biology in 2021; ^#^ calculated based on the average USD exchange rate during 2021.

## Data Availability

The cycle threshold data used in this study are available as a file at the Applied Biotechnology Laboratory of the Hospital das Clínicas at the Botucatu Medical School; The vaccination status is not public and was obtained from the E-sus, SIVEP-Gripe, and Vacivida systems.

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
