# Peer review of "Saliva as a Biological Fluid in SARS-CoV-2 Detection"

_diagnostics, 2024, doi:10.3390/diagnostics14090922_

Round 1

Reviewer 1 Report

Comments and Suggestions for Authors

Overall presentation of results is fine. However, manuscript needs further improvement.  

1.The results need more better interpretation to improve manuscript quality. It is suggested to discuss how virus enters body and what is its possible distribution in body fluids and explain your results accordingly.  

2. Provide details for viral RNA extraction.

3. Figure 1, labels on Y axis are not visible. 

4. Use better quality figures. Adding graphical abstract would improve manuscript quality. 

5. Manuscript language needs proofreading. 

Comments on the Quality of English Language

Language needs proofreading and minor editing. 

Author Response

1.The results need more better interpretation to improve manuscript quality. It is suggested to discuss how virus enters body and what is its possible distribution in body fluids and explain your results accordingly.

Authors: We inserted a paragraph explaining how the virus infects cells. Please, see lines 239-250.

2. Provide details for viral RNA extraction.

Authors: The text was properly modified. Please, see lines 87-95.

3. Figure 1, labels on Y axis are not visible.

Authors: We modified the figure. Please, see figure 2.

4. Use better quality figures. Adding graphical abstract would improve manuscript quality.

Authors: We modified the figures. Please, see figures 1, 2 and 3.

5. Manuscript language needs proofreading.

Authors: We revised the full manuscript and made several modifications. We also check grammar using automatic checkers such as Grammarly and Writefull.

Reviewer 2 Report

Comments and Suggestions for Authors

 I find this manuscript well written. Authors reported that  RT-PCR on saliva collection as a feasible alternative to the diagnosis of COVID-19. A saliva sample is also more comfortable  than a nasal swab. I suggest to edit this sentence Nasiri & Dimitrova and Roque and colleagues and to report near authors et al Nasiri et al and..Roque et al

Author Response

I find this manuscript well written. Authors reported that RT-PCR on saliva collection as a feasible alternative to the diagnosis of COVID-19. A saliva sample is also more comfortable than a nasal swab. I suggest to edit this sentence Nasiri & Dimitrova and Roque and colleagues and to report near authors et al Nasiri et al and..Roque et al

Authors: We modified this part properly. Please, see lines 45-46.

Round 2

Reviewer 1 Report

Comments and Suggestions for Authors

Revised version is much improved.